# The Impact of Premortality Stress on Some Quality Parameters of Roe Deer, Wild Boar, and Red Deer Meat

**DOI:** 10.3390/foods11091275

**Published:** 2022-04-28

**Authors:** Kristijan Tomljanović, Marijan Grubešić, Helga Medić, Hubert Potočnik, Tomislav Topolovčan, Nikolina Kelava Ugarković, Nives Marušić Radovčić

**Affiliations:** 1Faculty of Forestry and Wood Technology, Univesity of Zagreb, Svetošimunska cesta 23, 10000 Zagreb, Croatia; ktomljanovic@sumfak.hr (K.T.); mgrubesic@sumfak.hr (M.G.); tomi.topolovcan@gmail.com (T.T.); 2Faculty of Food Technology and Biotechnology, Univesity of Zagreb, Pierottijeva 6, 10000 Zagreb, Croatia; hmedic@pbf.hr; 3Biotehniška Fakulteta, Univerza v Ljubljani, Jamnikarijeva 101, 1000 Ljubljana, Slovenia; hubert.potocnik@gmail.com; 4Faculty of Agriculture, Univesity of Zagreb, Svetošimunska cesta 25, 10000 Zagreb, Croatia; nkelava@agr.hr

**Keywords:** free living animal, game meat, premortal stress level, red deer, wild boar, roe deer, pH, WHC, water content, meat colour L*, a*, b*

## Abstract

The specifics of meat production from free-ranging animals include the killing of animals in the wild with firearms. This type of uncontrolled killing sometimes leads to the phenomenon that the game does not die immediately but after a certain time from the shot to death, which may ultimately affect the quality of the meat. During one hunting year on free-ranging red deer (*Cervus elaphus*) (RD), roe deer (*Capreolus capreolus*) (RoD), and wild boar (*Sus scrofa*) (WB), the effect of time from shot to death on final pH, water-holding capacity (WHC), water content, and colour (L*, a*, b*) was investigated. All analyses were performed on *Musculus biceps femoris* (BF). After shooting, the animals were divided into two categories (A = time from shot to death ≤ 1 min; B = time from shot to death > 1 min). In RD, group B had significantly lower (*p* < 0.05) water content. In RoD, group B had significantly lower (*p* < 0.05) values of L* and b*. In WB, group B had significantly lower (*p* < 0.05) L* value and significantly higher (*p* < 0.05) pH value. The study proves that in BF of the three studied game species, the time extension from shot to death significantly affects the final water content values in RD, L* and b* in RoD and pH and L * in WB.

## 1. Introduction

Recently, much attention has been paid to organic food production, with controlled use of artificially produced nutrients that promote animal growth and reproduction, and the production of larger quantities of meat in a shorter time and in a smaller space. Game raised in open hunting grounds is a prime example of a food source that has minimal human impact and high nutritional value [1,2]. In the wild, however, game freely chooses its direction of movement and migration, dwelling areas, and food sources [3]. Due to the decreased diet and movement control, it is susceptible and prone to other habitat factors, including pathogenic microorganisms [4,5,6,7,8,9,10,11,12,13,14,15] and toxic elements, especially heavy metals [16,17,18,19], due to reduced food and movement control. Because they are free-ranging animals, they are also killed in their habitat, which makes manipulation of the meat difficult and increases the likelihood of unwanted contamination of the meat [20,21,22,23], especially during the warmer season when the likelihood of postmortem contamination increases significantly [24,25]. Compared to the meat of related/comparable domestic animals, the meat of free-ranging wild animals usually has lower fat content [26,27], darker colour [28,29], more favourable fatty acid composition [30], and mostly higher water content, which depends on several factors [31,32,33,34]. The meat of wild ruminants and wild boars is somewhat firmer compared to that of domestic pigs or ruminants [35], which is due to the structure of muscle tissue, and, sometimes, it has higher pH and unstable surface colour [36,37]. The water-holding capacity (WHC) of wild biungulate animals varies widely and depends on a variety of factors that have been investigated in studies [38,39,40,41,42]. Considering only the sensory factors, wild meat may be perceived as less juicy, firmer, and drier due to its lower intramuscular fat content, higher pH at the time of hunting, greater water-holding capacity, and the structure of the muscle tissue itself [39,43,44,45].

In addition to genetic inheritance, sex, age, and diet, postmortem treatment of meat also affects the meat quality of certain species of free-ranging animals [46,47,48]. Studies [49] have shown that the meat of fallow deer (***Dama dama***) suspended by the pelvic bone during evisceration has different characteristics than the meat of deer suspended by the Achilles tendon during processing. In addition, it has been found that the time of day when the game is hunted, the time of evisceration, or the hunting season also have an influence on the quality and maturation process of game meat [50]. In general, the qualitative characteristics of game meat are most influenced by lifestyle, i.e., diet, in addition to hereditary characteristics. Since red deer (***Cervus elaphus***) (RD), roe deer (***Capreolus capreolus***) (RoD), and wild boar (***Sus scrofa***) (WB) live in natural habitats and feed seasonally, there is considerable inter- and intraspecific variability in all qualitative parameters [23].

As mentioned earlier, one of the peculiarities, i.e., problems in the production/extraction of meat from wild animals is the killing method, which is very different from that used for domestic animals. While domestic animals are killed in slaughterhouses, free-ranging animals are killed in the wild, mainly with the use of firearms. This method of killing, which is usually not instantaneous, may have some effect on certain parameters of meat quality, especially pH, colour, water content, and WHC [23]. The stress caused by the shot during the period from wounding to death leads to the accumulation of glycogen in the intercellular space. At the time of the animal’s death, the increased amount of undissolved glycogen forms the basis for lactic acid production, and the pH of the meat decreases [51,52], resulting in a darker colour and increased water-holding capacity in the subsequent meat maturation process [53]. In some cases, game meat can have very high pH values, resulting in dark, firm, and dry meat (DFD), which has negative characteristics from the end consumer’s point of view [54].

To determine the effect of ante-mortem stress on water content, colour, WHC, and pH, shot individuals from RD, RoD, WB were processed within one calendar year for this study. The objective of the study was to compare the above parameters between two groups of animals where the time from shooting to death is different.

## 2. Materials and Methods

### 2.1. Animals

A total of 52 WB, 45 RoD, and 51 RD individuals were processed. Although the plan identified an equal number of males (M) and females (F), the field conditions and availability of sampling areas determined the final sex ratio structure, which is shown in Table 1. The study was partially conducted in the Republic of Croatia in Dinarides area, which is home to large predators (grey wolf (*Canis lupus*), European lynx (*Lynx lynx*), and brown bear (*Ursus arctos*)) that feed on large herbivores and wild boar to meet their food requirements [55], which is why hunters often avoid hunting females to preserve and maintain the annual birth ratio. Immediately after the animals were killed and initial measurements were taken, they were transported to game meat processing plants where they were eviscerated.

### 2.2. Data Collection

During the period of one year, samples of RD, RoD, and WB were collected in the western and central parts of the Republic of Croatia (Figure 1) during the planned hunt [56]. The animals were sorted by species, sex, and hunting type (selective/individual hunt (SH) and drive hunt (DH)). Selective or individual hunt is a type of hunting in which game is targeted at close range. The game is shot in its natural habitat, usually in open pastures at dawn or dusk, and the game is not disturbed before the hunt. Drive hunt is a type of hunting with dogs that is practised in central Europe. In this type of hunting, the game is chased by the hunters (driven hunting), and the dogs chase the game towards the hunters. This means that the game shot in this type of hunting is always excited and under great stress [23]. According to Croatian law, drive hunt is allowed only for WB, so some of the specimens were also classified in this category (Table 1). The study was conducted in seven hunting areas. Four of them are mountain and highland types (average altitude above 200 m), while the other three are lowland-type hunting grounds (average altitude below 200 m). The mountain hunting grounds are located in the Gorski Kotar region in Dinarids (45°24’04.62” N; 14°48’01.75” E). This is an area in the western part of Croatia with a rugged terrain of varying exposure. The altitudes range from 200 to 1500 m.a.s.l. As far as vegetation is concerned, fir (*Abies alba*) and spruce (*Picea abies*) forests are the most widespread. According to the Köppen classification, it is an area with boreal climate. The temperature of the coldest month of the year is between −3 and +18 °C. The driest time of the year is summer, when the average daily temperatures are above 23 °C. The highest amounts of precipitation fall in early spring and autumn. Snow lasts for an average of three months [57]. The hunting grounds in the lowlands are located in central Croatia (45°23’53.68” N; 16°53’40.62” E). The average altitude is between 100 and 200 m.a.s.l. There is a typical continental climate with all four seasons. The average annual air temperature is 10.3 °C. The average temperature during the growing season is 16.2 °C. The warmest month is July, the coldest is January. The absolute minimum is −23.0 °C, and the absolute maximum is 39.0 °C. Late frost in May is characteristic for this area. The average annual precipitation is 915 mm. As for vegetation, the most remarkable plant communities are pedunculate oak forests (*Quercus roboris*) and narrow-leaved ash forests (*Fraxinus angustifolia*) [58].

### 2.3. Field Measurement

After shooting, the time elapsed from shooting to death was measured for each individual specimen (respiratory reflex arrest), and the sex was noted (M/F). The time elapsed from shooting to death was measured using a traceable digital stopwatch. Since all animals were shot using a firearm (hunting rifle) from a distance, a certain period of time following the shot was necessary to reach the animal and ascertain its death. In this study, the specimens were classified into group A = time from shot to death < 1 min; and B = time from shot to death ≥ 1 min. A field scale (Kern CH 200 K 100) was used to measure body weight with the precision of 100 g. The *M. biceps femoris* (BF) was taken from each specimen. The BF sample was stored in a portable cooling chamber at +4 °C and taken to the laboratory in a portable cooler. After being shot, each animal was marked using an ear tag and transported for further processing.

### 2.4. Physicochemical Analysis

Water content was determined using the official method [59]. For pH measurement, a suspension of 10 g of the sample was mixed with 40 mL distilled water, and pH was measured by pH meter 24 h postmortem (benchtop sensION tm + MM374, Hach Company, Loveland, CO, USA).

For the determination of water-holding capacity (WHC), the filter paper press method was used to measure the amount of water expressed from a sample kept under pressure. In short, 2 g of BF muscle tissue was placed on a Whatman no.1 filter paper (which was weighted previously) and pressed between two glass plates by a weight of 200 g for 15 min. The water, which was squeezed out, was absorbed by the filter paper. WHC is obtained by the difference of the weight of the filter papers after and before measurements and is related to the weight of the sample. In this method, the amount of water expressed is inversely proportional to WHC [60]. Each sample was analysed in five replicates.

### 2.5. Colour Parameters

Colour measurements were carried out with a Minolta CM-700d (Osaka, Japan) spectrophotometer (illuminant D65, 10° standard observers, 8 mm aperture, with open cone). The L* (lightness), a* (redness), and b* (yellowness) colour was measured [61]. Before analysis, the spectrophotometer was calibrated with White Calibration Cap CM-A177. Each sample was analysed in six replicates.

### 2.6. Statistical Analysis

Statistical analysis was performed by the Statistica 14.0.0.15, TIBCO software Inc. (Palo Alto, CA, USA). One-way ANOVA was used to analyse data with normal distribution. The Shapiro–Wilk test was used for variable homogeneity test. A 5% significance level was used for determining significant differences in body mass, colour, and physicochemical meat composition. The Pearson linear correlation coefficients were used for checking the water content, WHC, pH, and colour L*, a*, b* correlation. Only significant correlations are presented in Section 3. The cartographic representation was performed in QGIS.

## 3. Results

A separate analysis of the animals killed in SH and DH was performed for WB. There was a significantly lower (*p* < 0.05) L* colour value in group B females shot in SH, indicating that the meat was significantly darker in females when the time from shot to death was extended (Table 2). Additionally, the females in group B had lower marginal difference in the colour indicator b* (*p* = 0.060).

With WB, some of the animals were killed in DH (Table 3). In DH, the pH values were significantly higher in female group B (*p* < 0.05). No regularity was observed with respect to WHC value, colour a*, and water content in both types of hunting. Due to the small sample, in female group B in SH (*n* = 3) and male group A in DH (*n* = 2) the differences obtained should be taken conditionally.

With RD, regularities were observed for pH, WHC, colour indicator L*, and water content (Table 4). Despite the fact that the differences are not significant, lower pH and L* values and higher water contents and WHC were observed in both sexes in group B. In females, there was a significant difference in water content between the observed groups (*p* < 0.05); the animals in group A had higher values.

In RoD, a significant difference was found between the observed groups in colour indicators L* and b* in females, whose mean values were significantly lower in group B (*p* < 0.05) (Table 5). The above colour indicators, although not significant, were also lower in males. In water content, there was a marginal tendency (*p* = 0.090) to show lower values in females in group B. The differences of the observed groups were not determined for pH, WHC, and colour indicator a*. Similar to some cases in WB and RD, there are a small number of specimens in group B females (*n* = 3), so the differences obtained should be taken conditionally.

Correlation analyses of the studied parameters were performed within each species separately for each sex. A positive correlation was found in WB females for pH value and colour parameter a* (r = 0.521; *p* = 0.032), and a negative correlation for water content and colour parameter b* (r = −0.507; *p* = 0.038). In female RD, we found a negative correlation for pH and WHC (r = −0.490; *p* = 0.039), and a positive correlation for water content and colour parameter L* (r = 0.478; *p* = 0.045). In male RoD, we found a positive correlation for pH value and colour parameter b* (r = 0.455; *p* = 0.020).

## 4. Discussion

The tradition of hunting and using large game meat is widespread in Europe, especially in countries in the centre of the continent (Austria, Poland, Hungary, and Germany) [23]. However, these countries also use their big game meat resources at the local level, and consumption is mainly limited to hunters and their families [20]. In Spain, consumption of deer meat is slightly more common, while in Scotland and England, small game is common. New Zealand has a 50% share in the world production of deer meat, the USA and Australia in the production of wild boar meat, Canada in the production of fallow deer meat, and Argentina in the production of field hare meat [62]. In Germany, the consumption of game meat in the total amount of meat consumed per capita is 0.9% [20]. In Italy, the consumption of wild boar, deer, and roe deer is 0.1–0.3 kg per capita, considering the total population, while in hunter families, it is 1–4 kg per person per year [23]. In general, it can be said that the consumption, marketing, and market of game meat in Europe are poorly developed [63,64]. Apart from being a relatively exotic food, there are many difficulties in the production of game meat. Industrial production requires uniformity of individual quality parameters, which is often not the case for game killed in the wild. Some variation in quality parameters is due to differences in dietary habits but also to the method of killing and stress, which is difficult to control, and the effects of which are difficult to predict. Prior to killing, game meat is subjected to great stress, which has a significant impact on meat quality via effects on muscle energy metabolism. This includes changes in metabolite concentration and glycogen content [65].

Game in the wild is usually killed with firearms, and this study has shown that in some cases, more or less time elapses from shooting to death, and it highlighted the cumulative effects of this stress on the meat. The stress that occurs during the slaughter of domestic animals or the shooting of wild animals directly affects pH discrepancies, which in turn affect WHC and colour [53]. This is due to the accumulation of larger amounts of glycogen in the intercellular space at the moment immediately preceding death. The undissolved glycogen forms the basis for lactic acid production after death, and pH decreases [51,52], leading to denaturation of proteins that bind free water, which also results in a lower WHC [49]. This phenomenon is more pronounced in warmer areas [66]. The higher ambient temperatures common to game hunting in spring, summer, and early fall further decrease pH, denature proteins, and lower WHC, ultimately resulting in pale meat with poor texture [39,44,45,67].

Correlations between colour parameters and WHC can be explained by variations in pH affecting both colour and WHC. The results of correlation analysis showed a positive correlation for pH and colour indicator a*, and a negative correlation for water content and colour indicator b* in WB females. In RD females, a negative correlation was found for pH and WHC, and a positive one for water content and colour indicator L*, while in male RoD, a positive correlation was found for pH and colour parameter b*.

The water content depends on the WHC. Water in muscle is entrapped in the structures of the cell, including the intra- and extra-myofibrillar spaces; therefore, significant changes in the structure of the cell affect the WHC. Protein degradation also plays a role in determining WHC. The pH directly affects the ability of myofibrillar proteins, myofibrils, and muscle cells to trap water. An accelerated pH drop and a low final pH are related to the development of a low WHC. A rapid pH drop while the muscle is still warm, causes denaturation of many proteins, including those involved in binding water [68].

Colour is influenced by the content and physicochemical state of myoglobin and by meat structure. Meat structure is directly related to the ultimate pH: at a high ultimate pH, muscle fibres are more densely packed together due to the increased water-holding capacity of the muscle protein. As a result, light is less scattered on the surface, and the meat appears darker. The relationship between meat colour and pH is generally accepted. L* is a reliable indicator of PSE and/or DFD pork and can be used as a measure of pork quality [69]. In addition, b* has been found to be selected as the best variable for detecting pH differences. The rate of postmortem pH decline affects biochemical reactions and structural properties of muscles and may also affect colour. Redness (a*) is generally more influenced by early postmortem pH. In pork, the combination of low early postmortem thigh muscle temperature leads to protein denaturation, resulting in early inactivation of oxygen-consuming enzymes. This promotes the formation of MbO2, leading to higher a* values. In deer, meat with higher ultimate pH was redder. Ultimate pH was further correlated with WHC [70].

### 4.1. pH Value

Final meat pH is the result of the amount of glycogen present in muscle prior to killing, which is highly dependent on factors responsible for physical and psychological stress. Exposure to stressors during the killing results in ATP reduction, which leads to depletion of muscle glycogen concentration. Significant depletion of muscle glycogen reserves prior to killing leads to higher final pH, which in turn has profound effects on several important meat quality traits [71]. A faster heart rate or higher catecholamine level before killing correlates with a high rate of early postmortem pH decline and a lower early postmortem pH, which affects meat colour and water-holding capacity. The higher the heart rate in the minutes before slaughter, the faster the rate of muscle pH drop in the early postmortem period [65].

There are a relatively large number of studies that have focused on the study of game meat [72,73,74,75,76,77,78,79,80], but only a few of them [81,82] address the effects of ante-mortem stress on pH, WHC, or colour. Considering only the sex, females showed higher average pH values. In the observed groups, pH was significantly higher (*p* < 0.05) in females killed in DH in group B, while in SH, the difference was not significant, although it was higher on average in group B. Studies conducted in WB show different values for the final pH value, as well as an ultimately high pH value. For example, studies [43,79] define an ultimately high pH as ≥ 5.8, while other studies [83] define it as pH ≥ 5.9, while, for example, some studies [84] on WB consider a pH ≥ 6.0 to be ultimately high in WB. Studies [85] for *M. semimembranosus* (SM) WB give a pH range of 5.45–5.88. The pH values determined in our study for group B in SH and DH are in the middle; however, in both cases, the final pH values are high, especially for female specimens. In animal group A, lower average values were obtained for both sexes, which can be considered normal [79]. In contrast to WB, higher average pH values were determined for both sexes in group A at RD. Studies [86] conducted in farmed female RD showed an average pH of 5.58 for BF. Comparing these data with the results of our study, it can be concluded that the pH of wild animals hunted with firearms is on average higher than the pH of animals slaughtered in slaughterhouses. Without taking diet into account, there are two possible explanations for these results. Animals raised in a controlled environment on a farm, without regard for attention and minimal stress, experience higher levels of stress prior to death, resulting in a lower final pH [51,52]. The other explanation is related to the higher stress and acceleration of cellular metabolism after shooting specimens hunted in the wild, which could lead to a later consumption of glycogen that is not compensated by the time of death. The lack of glycogen results in lower lactic acid production and higher final pH [53]. Since RD is hunted in single hunts, and the animals were not subjected to extreme exertion prior to shooting, it is possible that the stress level that occurs in a controlled environment prior to slaughter is cumulatively higher than that which occurs at the moment of shooting from a firearm, which is understandable due to the fact that wildlife is involved. The confirmation of this thesis is found in the studies of two groups of RD [87], in which the authors found low average plasma cortisol concentrations (less than 7 ng/mL) and a muscle pH less than 5.74 in the animals killed in the wild, while the RD animals kept on a farm and transported to the slaughterhouse had high cortisol concentrations greater than 20 ng/mL and muscle pH greater than 5.74. Studies at a RD farm [88] show a pH of 5.44 in males and a pH of 5.45 in female animals slaughtered at a slaughterhouse, again, significantly lower than the values in this study. In RoD, group B animals had lower average pH values in both sexes. In males, the difference, although not significant, is present, while in females, it is insignificant *p* = 0.986. Studies [89] for the *M. longissimus dorsi* (LD) found pH values of 5.48 in females and 5.47 in males. Studies [90] performed in Lithuania on adult RoD specimens shot in the neck or head area, implying rapid death, gave a pH of 5.44. Studies [91] performed in two regions of the Czech Republic on *M. gluteus medius* (GM) gave a pH of 5.50 for one region and 5.65 for the other region. Studies [92] conducted in the subalpine region of Austria reported a LD pH, depending on the hunting season, ranging from pH 5.58 in the autumn to pH 5.66 in the spring. In relation to the recorded data, this study obtained relatively high values for the final pH, which are closest to the values obtained during the spring hunting season in Austria [85]. In the above study, the temperature is not mentioned, but it is possible that this very variable has an influence on pH values in the autumn and spring. Although not explicitly noted, this variable could also have had a specific influence on the final pH of RoD in our study, as the majority of animals were killed in the warmer season.

### 4.2. Water-Holding Capacity (WHC)

The WHC value of meat is important to maintain its juiciness, and increasing its value improves its sensory properties. Many factors, such as pH and postmortem proteolysis, influence WHC by altering the amount and location of moisture in muscle. A rapid decrease in pH combined with high muscle temperature in the early postmortem period leads to denaturation of approximately 20% of muscle proteins, resulting in a loss of their functionality and ability to bind water. The WHC value is closely related to the colour of the meat and also influences other physical properties, including meat texture [93]. In our study at WB, no significant difference was found in WHC relative to hunting type. In general, there are few studies addressing the effects of premortem stress on WHC in game meat. Research in WB [94], conducted in Poland on three different weight classes, shows a LD WHC ranging from 10.43% to 11.76%. Interestingly, the lowest WHC was found in animals with medium weight, while lighter and heavier animals had higher WHC values, on average. Studies in WB [95], performed on animals with body weights ranging from 38.6 to 45.9 kg, found LD WHC 9.91% in females and a significantly higher value of 12.43% in males. In contrast to WB, this study showed higher average WHC values in RD in group A. In females, the median difference was slightly smaller. In RoD, the values were similar to those in RD, with higher average WHC values recorded in group A. The values were slightly higher in females than in males. In general, the WB animals in group B had higher average values, while in RD and RoD, the case was reversed, i.e., higher average values were recorded in group A. This situation could be explained by the direct relationship between pH, WHC, and temperature at the time of the shooting. In contrast to WB, which was mainly hunted in winter, RD and RoD were mainly hunted in spring, summer, and autumn, when outdoor temperatures are relatively high. Due to the sudden drop in the pH of meat in a warm environment, the muscle proteins denature, and some of the bound water is released [39.43–45.66%]. Possibly, this is one of the reasons why the specimens of group A, which do not have a sudden pH drop compared to group B, have higher WHC values. In the case of wild boar, the situation is reversed. The meat of group B had higher average WHC values. However, the majority of WB specimens were killed in the colder season, and the stress to which the animals are subjected due to wounding and the long time before death leads to increased glycogen consumption, the value of which remains low after death. Since glycogen is critical for the production of lactic acid in muscles, its decreased content leads to lower lactic acid production, resulting in a higher final pH. A high pH stops the denaturation of muscle protein, and under these conditions, the meat has a higher water-holding capacity than the group A specimens, in which a normal glycogen level resulted in the production of more lactic acid, a slightly lower pH, partial denaturation of some of the protein, and finally, a lower WHC. The meat of the specimens with a high final pH is essentially dry, firm, dark meat (DFD) and has poor sensory characteristics compared to meat with normal colour and texture, despite the increased water-holding capacity [96].

### 4.3. Water Content

In WB, no significant difference in the water content of the studied groups was found for both observed hunting types. With the exception of DH females, the recorded values are slightly higher in group B. In RD, water contents were consistent with the trend described for WB, and the averages for both sexes were higher in group B. In group B, the determined value for female specimens was significantly higher than the value in group A. For both WB and RD, with the described exception, water contents show some regularity and are on average higher in group B than in group A for both sexes and both hunting types. It can be concluded that as the time interval between shooting and death increases, the percent water content also becomes higher. For RoD, the case is reversed. In group A, higher average water contents were found in both sexes. There are no published studies with which to compare the values obtained by this experiment. Some results, such as Ref [85], show water contents in juvenile WB of 75.96% in males and 76.64% in females. In subadults, the authors give results of 75.84% in males and 74.96% in females, which is in line with the results obtained in our study.

### 4.4. Meat Colour (L*, a*, b*)

The colour of fresh meat is also significantly affected by postmortem metabolism. An abnormally low pH causes denaturation of muscle proteins, including myoglobin, and reduces their ability to bind water. As a result, large amounts of water migrate from the interior of muscle fibres into the extracellular space, increasing light reflection and resulting in a paler meat colour. In contrast, high pH makes flesh colour appear darker because WHC is higher and light absorption, reflection, and protein denaturation are lower. High pH also promotes the activity of oxygen-depleting enzymes, resulting in less oxygen available to bind myoglobin and more deoxymyoglobin being formed [93].

For female WB shot at SH, the average L* value for BF in group A was significantly lower (*p* < 0.05) compared to group B. In general, WB males had slightly lower L* values than females. In males, there is also a regularity in colour indicators a* (red) and b* (yellow) in DH and SH, with both being lower on average in group B. For females hunted in SH, there is almost no difference in a* value in the observed groups (*p* = 0.979), while b* value is lower in group B, on average. Studies [73] in WB, conducted in Brazil with MLD, showed higher L* and b* values and slightly lower a* (L* = 51.30, a* = 7.94, b* = 13.24) compared to our study. Similar LD results were found in a study [74] conducted in Poland on female WB of different weight classes, in which a regularity in the decrease in L* value (lightness) with increasing weight was found, directly related to age to some extent. At RD, there were no significant differences between the observed groups; however, the determined values of all three colour indicators (L*, a*, b*), with the exception of b* in females, were, on average, lower in group B. The values determined for RD are slightly higher than those for *M. triceps brachii* (MTB) of Ref. [97] and significantly lower than those for *M. longissimus thoracis et lumborum* (LTL) of Ref. [98]. An almost identical representation to RD in terms of colour indicators was found in RoD—all indicators except b* in female specimens had lower mean values in animals in group B. In RoD, there was a significant difference between L* and b* values in females. Both indicators had significantly lower values (*p* < 0.05) in group B. Of the three species observed, males RD had the lowest L* values, while the highest values were found in WB females hunted in DH. WB meat stands out with the highest average b* values (yellow), which could indicate a higher content of intermuscular fat in contrast to the other two species observed. According to Ref. [54], game meat is generally darker than domestic animal meat due to diet and lifestyle, as well as some stress during hunting, which can sometimes lead to a negative attitude towards this type of food among the end consumers. Our study shows and demonstrates an important regularity that could complement this statement in the future. In all three species and both sexes, lower average L* (lightness) values were found in group B, implying that the increase in time from shooting to death causes the meat to take on a slightly darker hue.

## 5. Conclusions

A particularly significant influence of the time elapsed from firing to death was found in water content, colour parameters L*, b*, and pH values. Interestingly, the significant differences mentioned above were found in females. Some regularity was observed in pH and colour parameter L* in both sexes. Considering each observed group, higher average pH values were found in group B in WB, while in RD and RoD, the opposite was true, i.e., animals in group B had lower average pH values. The colour parameter L* had lower average values in all three species studied in group B, implying that increasing the time elapsed from shooting to death results in darker meat. This study demonstrates certain regularities and somewhat describes the effects of increased time between shooting and death on pH, meat colour, and water content. However, this study is limited. Future analyses will need to consider a larger number of specimens and add variables, such as outside temperature at the time of shooting, level of residual blood and loss of blood, age, diet, etc., to determine their potential effects on some of the observed indicators related to ante-mortem stress and its cumulative effects on meat quality.

## Figures and Tables

**Figure 1 foods-11-01275-f001:**
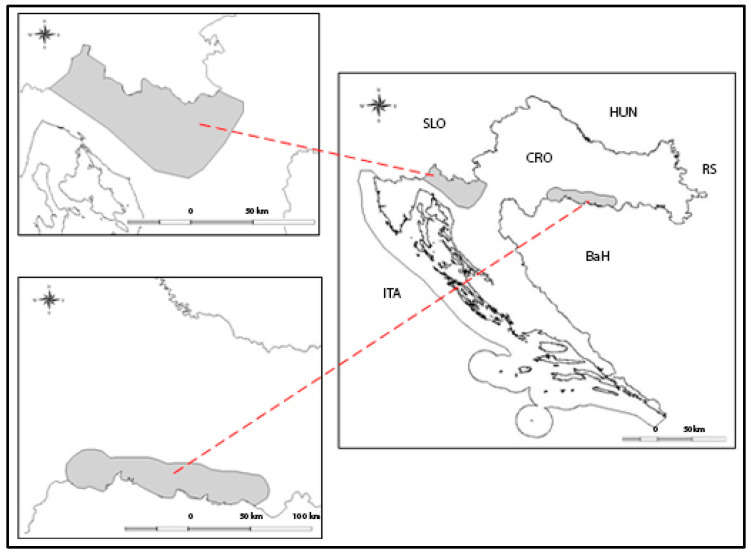
Gorski Kotar in north-west part of Croatia (**top left**). Lowland in central part of Croatia (**bottom left**). Overview map (**right**).

**Table 1 foods-11-01275-t001:** Body weight (mean ± SE) of the red deer (RD), roe deer (RoD), and wild boar (WB), as influenced by time from shot to death (A ≤ 1 min; B > 1 min).

Species	Sex									
	Male				*p*-Value	Female				*p*-Value
	A	*n*	B	*n*		A	*n*	B	*n*	
RD	140.65 ± 13.19	16	172 ± 14.09	14	0.112	105.65 ± 9.11	14	97.86 ± 12.88	7	0.627
RoD	23.71 ± 1.25	14	24.40 ± 1.09	18	0.683	22.25 ± 1.87	10	28.33 ± 3.42	3	0.147
WB (SH)	64.45 ± 7.22	9	53.80 ± 6.85	10	0.299	52.98 ± 11.81	6	63.37 ± 16.70	3	0.627
WB (DH)	83.67 ± 11.77	7	73.70 ± 22.03	2	0.701	59.62 ± 9.08	8	55.74 ± 9.71	7	0.775

WB (SH) = wild boar shot in selective/individual hunt, WB (DH) = wild boar shot in drive hunt.

**Table 2 foods-11-01275-t002:** pH value, WHC, colour (L*, a*, b*), and water content for *M. biceps femoris* of wild boar (WB) shot in a selective hunt (SH) (mean ± SE).

Parameter	Sex					
	Male		*p*-Value	Female		*p*-Value
	A (*n* = 9)	B (*n* = 10)		A (*n* = 6)	B (*n* = 3)	
pH	5.64 ± 0.04	5.72 ± 0.04	0.170	5.63 ± 0.04	5.67 ± 0.07	0.583
WHC (%)	11.36 ± 1.32	13.95 ± 1.25	0.173	12.70 ± 1.36	16.97 ± 1.93	0.113
L*	35.80 ± 1.35	34.92 ± 1.28	0.645	39.98 ± 1.19	32.08 ± 1.69	0.006
a*	11.03 ± 1.37	9.09 ± 1.37	0.319	10.46 ± 2.49	10.58 ± 3.53	0.979
b*	11.26 ± 0.82	10.56 ± 0.78	0.544	13.11 ± 0,55	10.99 ± 0.77	0.060
Water (%)	75.33 ± 0.40	76.24 ± 0.38	0.115	75.47 ± 0.63	76.72 ± 0.89	0.293

**Table 3 foods-11-01275-t003:** pH value, WHC, colour (L*, a*, b*), and water content for *M. biceps femoris* of wild boar (WB) shot in a drive hunt (DH) (mean ± SE).

Parameter	Sex					
	Male		*p*-Value	Female		*p*-Value
	A (*n* = 7)	B (*n* = 2)		A (*n* = 8)	B (*n* = 7)	
pH	5.69 ± 0.04	5.72 ± 0.08	0.738	5.61 ± 0.06	5.86 ± 0.05	0.012
WHC (%)	12.25 ± 1.36	11.53 ± 2.54	0.810	11.42 ± 1.05	11.82 ± 1.12	0.799
L*	37.81 ± 1.47	35.72 ± 2.75	0.524	40.01 ± 1.27	38.52 ± 1.35	0.435
a*	18.26 ± 2.41	15.85 ± 4.50	0.653	16.52 ± 1.39	19.00 ± 1.48	0.243
b*	13.45 ± 1.11	13.34 ± 2.07	0.963	13.29 ± 0,68	12.40 ± 0.72	0.386
Water (%)	75.23 ± 0.41	75.52 ± 0.76	0.753	74.67 ± 0.37	75.24 ± 0.40	0.313

**Table 4 foods-11-01275-t004:** pH value, WHC, colour (L*, a*, b*), and water content for *M. biceps femoris* of red deer (RD) (mean ± SE).

Parameter	Sex					
	Male		*p*-Value	Female		*p*-Value
	A (*n* = 14)	B (*n* = 16)		A (*n* = 14)	B (*n* = 7)	
pH	5.75 ± 0.03	5.68 ± 0.03	0.169	5.81 ± 0.04	5.78 ± 0.05	0.604
WHC (%)	10.65 ± 0.81	12.29 ± 0.81	0.164	9.64 ± 0.87	11.18 ± 1.28	0.333
L*	31.07 ± 0.92	30.05 ± 0.98	0.457	34.26 ± 1.23	33.99 ± 0.87	0.856
a*	13.35 ± 1.19	11.85 ± 1.28	0.395	15.86 ± 1.33	18.45 ± 1.88	0.274
b*	9.38 ± 0.43	9.66 ± 0.46	0.653	11.62 ± 0,71	10.83 ± 1.01	0.528
Water (%)	75.78 ± 0.27	75.89 ± 0.29	0.771	74.56 ± 0.27	75.62 ± 0.39	0.038

**Table 5 foods-11-01275-t005:** pH value, WHC, colour (L*, a*, b*), and water content for *M. biceps femoris* of roe deer (RoD) (mean ± SE).

Parameter	Sex					
	Male		*p*-Value	Female		*p*-Value
	A (*n* = 14)	B (*n* = 18)		A (*n* = 10)	B (*n* = 3)	
pH	5.70 ± 0.03	5.67 ± 0.03	0.350	5.86 ± 0.03	5.85 ± 0.05	0.986
WHC (%)	11.35 ± 0.90	9.72 ± 0.84	0.200	12.62 ± 0.75	11.38 ± 1.68	0.511
L*	33.38 ± 0.98	31.90 ± 0.87	0.268	36.71 ± 0.93	32.28 ± 1.70	0.043
a*	9.26 ± 0.59	9.27 ± 0.51	0.984	11.18 ± 0.69	12.84 ± 1.27	0.280
b*	10.45 ± 0.52	10.62 ± 0.45	0.799	10.93 ± 0.32	9.33 ± 0.59	0.037
Water (%)	74.85 ± 0.31	74.52 ± 0.28	0.448	73.84 ± 0.21	73.03 ± 0.38	0.090

## Data Availability

Data is contained within the article and also available on request.

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
