# Peer review of "The Impact of Premortality Stress on Some Quality Parameters of Roe Deer, Wild Boar, and Red Deer Meat"

_foods, 2022, doi:10.3390/foods11091275_

Round 1

Reviewer 1 Report

GENERAL COMMENT:

I consider this work is within the scope of “Foods”. It contains information useful in a field in which available information is of interest to improve game meat quality. Overall, the manuscript is well written and structured, and its revels expertise of the authors on the subject. I few proposals to be considered by the authors to improve the manuscript. I indicate these recommendations below and in a commented PDF file I have uploaded.

My main concerns relate to two facts:

  1. a) Sample size in several treatments in tables 2, 3 and 5 are two low. Therefore, it is necessary to indicate this fact in the Discussion and, therefore, to interpret these results in a prudent way and with exploratory scope.
  2. b) In several cases, in the Results section the authors describe differences between treatments when P>0.05. In my opinion this is wrong. It is acceptable, however, to highlight differences when there is a marginal tendency to show differences, when 0.05 < p < 0.1.

ABSTRACT:

See commented version for typos to be corrected.

Lines 16-17: Add scientific names of the game species.

KEYWORDS:

Line 26: Correct typo “freliving”.

INTRODUCTION:

Lines 55 and 61: Insert scientific names of game species.

MATERIALS AND METHODS:

Line 83: Improve sentence writing.

Line 86: Add country.

Line 87: Insert scientific name of animals.

Line 165: Correct typo: “homogenecy”.

RESULTS:

See commented version for typos to be corrected.

You can not describe differences between treatments when P>0.05. In my opinion this is wrong. It is acceptable, however, to highlight differences when there is a marginal tendency to show differences, when 0.05 < p < 0.1.

TABLES:

In some cases, sample size is low for several treatments (n=2, n=3). Take into account this limitation for the discussion.

Tables 2 an3, title: Insert comma: "L*, a*".

REFERENCES SECTION:

Check this section to correct several typos and to write in italics scientific names of organisms.

For example, do not mix long and short dashes to separate page numbers. See recent articles of the journal to use the same dash kind.

Revise for consistency journal names format: abbreviated of with full word (according to Instructions for authors and template).

Reviewer 2 Report

The manuscript should be reviewed by the authors or statistical editor for statistical tests applied and the associated interpretation of results.

The work presented for review is concerned with the assessment of the effect of ante-mortem stress on water content, color, water holding capacity and pH in Biceps femoris muscle of tree game animal species (wild boar, reed dear and roe dear). Authors compared the above mentioned parameters between the groups of animals where the time from shooting to death are different. The paper is written in a way that accurately describes the problem.

Comments:

Line 97-98 - Error in explanation of abbreviations

Line 162-167 – Authors should check and clarify which test was used for variable homogeneity test and which for normal distribution. Is there any post-hoc test used to find a significant difference between A and B group of animal

  1. An important parameter assessing the quality of game meat is incomplete loss of blood and level of residual blood. The lack of this parameter significantly reduces the value of this paper. IF the authors made this determination it is recommended to include this parameter in the manuscript.
  2. The English should be checked due to some minor errors.
  3. The list of references should be prepared according to guide for authors, in a uniform way, surnames and name abbreviations should be checked again by authors.
